# Invasive Infections Associated with the Use of Probiotics in Children: A Systematic Review

**DOI:** 10.3390/children8100924

**Published:** 2021-10-16

**Authors:** Martina D’Agostin, Domenica Squillaci, Marzia Lazzerini, Egidio Barbi, Lotte Wijers, Prisca Da Lozzo

**Affiliations:** 1Department of Medicine, Surgery and Health Sciences, University of Trieste, 34127 Trieste, Italy; martina.dagostin@gmail.com (M.D.); egidio.barbi@burlo.trieste.it (E.B.); priscadalozzo@gmail.com (P.D.L.); 2Institute of Maternal and Child Health—IRCCS Burlo Garofolo, Via dell’Istria 65/1, 34137 Trieste, Italy; marzia.lazzerini@burlo.trieste.it; 3Faculty of Health, Medicine and Life Sciences, Maastricht University, 6229 ER Maastricht, The Netherlands; lotte.wijers34@gmail.com

**Keywords:** probiotics, invasive infections, sepsis, preterm

## Abstract

Although the effectiveness of probiotics has only been proven in specific conditions, their use in children is massively widespread because of their perception as harmless products. Recent evidence raises concerns about probiotics’ safety, especially but not only in the paediatric population due to severe opportunistic infections after their use. This review aimed at summarising available case reports on invasive infections related to probiotics’ use in children. For this purpose, we assessed three electronic databases to identify papers describing paediatric patients with documented probiotic-derived invasive infections, with no language restrictions. A total of 49 case reports from 1995 to June 2021 were identified. The infections were caused by *Lactobacillus* spp. (35%), *Saccharomyces* spp. (29%), *Bifidobacterium* spp. (31%), *Bacillus clausii* (4%), and *Escherichia coli* (2%). Most (80%) patients were younger than 2 years old and sepsis was the most observed condition (69.4%). All the patients except one had at least one condition facilitating the development of invasive infection, with prematurity (55%) and intravenous catheter use (51%) being the most frequent. Three (6%) children died. Given the large use of probiotics, further studies aiming at evaluating the real incidence of probiotic-associated systemic infections are warranted.

## 1. Introduction

According to the revised definition of the International Scientific Association for Probiotics and Prebiotics (ISAPP), probiotics are “live microorganisms that when administered in adequate amounts, confer a health benefit on the host” [1]. During the last two decades, interest in probiotic supplements that modify the microbiota to confer health benefits has been growing, leading to the widespread use of many different types of probiotics in both community and healthcare settings.

At the current state of knowledge, health benefits have been demonstrated for specific probiotic strains of the following genera: *Lactobacillus, Bifidobacterium, Saccharomyces, Enterococcus, Pediococcus, Leuconostoc, Bacillus*, and *Escherichia coli*, which are either derived from the intestinal microbiota of healthy humans or from the fermentation of dairy products [1,2]. Probiotics have proven to be effective as a preventive intervention for a few specific conditions, such as neonatal necrotising enterocolitis (NEC) and late-onset sepsis (LOS) in premature infants [3,4,5,6]. Their benefit in conditions such as diarrhoea, allergic diseases, and infantile colic showed conflicting results and low levels of evidence [7,8,9,10,11,12].

Nevertheless, the probiotic’s market continues to grow rapidly worldwide, also due to the perception of their safety [13]. The general assumption that probiotics are safe derives from a long history of probiotic use and mixed data from clinical trials, and animal and in vitro studies [14]. However, only a few systematic safety studies have been carried out, particularly in vulnerable populations, and most of the existing published studies on probiotics have not specifically reported safety, leaving uncertainties regarding their potential risks [15,16]. In 2018, a systematic review of 384 randomised controlled trials assessing probiotics and prebiotics found that safety outcomes for these interventions in the literature were often missing, insufficient, or inconsistent; of the 384 trials, 106 (28%) did not provide any information related to harms, and 311 (81%) did not mention adverse events in the abstract [17].

Increasing evidence raises concerns about the probiotics’ opportunistic potential of causing bloodstream and deep-seated infections, especially in high-risk groups such as preterm neonates and critically ill patients [18,19]. A Cochrane review assessing the efficacy and safety of probiotics’ use in children with antibiotic-associated diarrhoea concluded that, although no serious adverse events were observed in the included studies, observational case reports and case review have reported serious adverse events in debilitated or immunocompromised children with underlying risk factors, including central venous catheter use and altered intestinal permeability [20].

Unlike adult patients, who are described to develop both systemic and deep-seated infections such as liver abscess [21], endocarditis [22,23], pleural empyema [24], and retropharyngeal abscess, children seem to develop primarily systemic bloodstream infections after probiotics’ administration [25].

This review aims to summarise the available evidence on the occurrence of invasive infections, such as sepsis, septic shock, bacteraemia, or pneumonia, associated with probiotics’ use in paediatric patients.

## 2. Materials and Methods

The review of the literature was limited to the paediatric population. All case reports and case series describing patients under 18 years of age with documented probiotic-derived invasive infections were included. The searches included papers from 1995 (first case reported) to 30 June 2021. No language restrictions were applied while selecting the papers. The following three electronic databases were searched: PubMed, Scielo, and Scopus. The following search string was used for PubMed: (“Probiotics”[Mesh] OR probiotic* OR Lactobacillus OR Bifidobacterium OR Saccharomyces) AND (“invasive infection” OR “adverse effects” OR sepsis OR bacteraemia OR fungemia) AND (“Child”[Mesh] OR paediatric* OR children). A combination of the above keywords was used for the other databases. Reference lists of all articles were manually searched for cross-references. Additional cases were identified from the references of the case reports. An attempt was made to obtain the full text of each case. When the full-text publication was unavailable via interlibrary loan, information involving the case was based on information in the article(s) that referenced the case report, if reported in sufficient details. Each article was selected and analysed for inclusion by two authors in parallel. The following data elements were extracted, from the selected studies, by two authors:−Publication data: title and authors of the article, name and year of the journal, volume, number, and pages of the articles.−Patient information: age, gender, underlying condition(s), exposure time to probiotics (in days), type and duration of the treatment (in days), and outcome (favourable or death).−Type of infection: identified microorganisms.

Invasive infection was defined as a severe infection in which the microorganism was isolated from a normally sterile site.

This review is reported according to Preferred Reporting Items for Systematic Reviews and Meta-Analyses (PRISMA) guidelines [26], but it was not pre-registered on The International Prospective Register of Systematic Reviews (PROSPERO).

## 3. Results

Out of 1537 studies, 49 documented cases of proven invasive infection in paediatric patients after probiotics’ use were identified [27,28,29,30,31,32,33,34,35,36,37,38,39,40,41,42,43,44,45,46,47,48,49,50,51,52,53,54,55,56,57,58,59,60,61,62].

Sepsis was the most observed condition, affecting 34 patients (69.4%). Fourteen patients (28.6%) presented bacteraemia or fungemia, while pneumonia was reported in one patient (2%). The probiotic microorganisms involved in the invasive infectious diseases and the details of each case are presented in Table 1, Table 2, Table 3 and Table 4.

We reported the denomination of the *Lactobacillus species* according to the original texts. However, we emphasize that the classification has been recently updated (Zheng et al., 2020). In light of the new classification, *Lactobacillus reuteri* is now called *Limosilactobacillus reuteri*, while *Lactobacillus rhamnosus* is called *Lacticaseibacillus rhamnosus*.

The infections were caused by *Lactobacillus* spp. in 17 patients (35%), *Saccharomyces* spp. in 14 patients (29%), *Bifidobacterium* spp. in 15 patients (31%), *Bacillus clausii* in 2 patients (4%), and *Escherichia coli* in 1 patient (2%). The length of probiotics’ intake varied largely, from 1 to 300 days. Probiotics were administered as enteral supplementation in all cases: orally or via a nasogastric tube or through gastrostomy, according to patients’ clinical conditions. In most reports, the etiological agents were identified both by blood cultures and strain-specific polymerase chain reaction (PCR) (61%) [27,28,29,30,31,32,33,35,42,50,51,52,53,54,56,58,59,62]; in 17 cases (34%), causative microorganisms were isolated from cultures [27,37,38,39,40,41,43,44,45,48,55,57,60,61], and in 3 of these (6%), the causative microorganisms were isolated both from blood culture and from central venous catheter (CVC) tip culture [46,47,49]. The pathogen was isolated from bronchoalveolar lavage (BAL) culture in the patient with pneumonia and confirmed with PCR analysis [36]. In one patient, the infection was confirmed with blood culture, CVC tip culture, and PCR analysis [34].

Most patients were younger than 2 years old. More specifically, 21 patients out of 49 (43%) were neonates (age < 1 month), while the other 18 (37%) were aged between 1 month and 2 years. Of the remaining patients, 4 (8%) were aged between 2 and 12 years, and only 2 (4%) between 12 and 18 years, while for 4 patients (8%), their age was not reported. The female/male ratio was 0.36 (16 female and 29 male). Males were more prone to develop invasive infections compared to females, with an OR of 3.01 (*p* = 0.08). Most of the patients (45/49) were receiving probiotics at the onset of the infection, with a median treatment duration of 10.0 days (IQR 1–21.5), while 4 cases did not take any probiotic supplements. For these patients, the invasive infection was reported to be caused by contaminations of probiotics administered to another child admitted in the same room.

All children except one had at least one condition facilitating the development of invasive infection, with prematurity and intravenous catheter use being the most frequently reported predisposing factors (55% and 51%, respectively). Gastrointestinal pathologies such as short bowel syndrome, enteral/parental nutrition, intestinal inflammation, and abdominal surgery were described in 23 cases (46.9%). Respiratory support, congenital heart disease, and genetic syndromes were present respectively in 9 (18.3%), 6 (12.2%), and 4 (8%) patients. Two patients were undergoing chemotherapy, one for acute lymphoblastic leukaemia, the other for acute myeloid leukaemia. Less frequent underlying conditions were cystic fibrosis, malnutrition, burns, immunosuppressive treatment, and renal failure.

The treatment of the invasive infections was specified for 41 patients. The median treatment duration was 12 days (IQR 10–14.5). Treatment drugs varied largely, including more frequently ceftriaxone, ampicillin, gentamicin, penicillin, ampicillin/sulbactam, vancomycin, and levofloxacin for bacterial infections and amphotericin B, fluconazole, and micafungin for fungal infections.

While most patients had a favourable outcome, three of them (6%) died. In two cases, the fatal outcome was considered to be related to multiple comorbidities rather than to probiotic infection itself.

Two of these were neonates taking probiotics according to a routine protocol for premature babies. One of the new-borns, a premature baby born at 27 weeks of gestational age, died after sepsis from *Lactobacillus reuteri* on the second day of life. The other new-born (27 weeks of gestational age) developed a *Saccharomyces boulardii*-related sepsis. The fungus appeared cleared after 72 h of antifungal therapy, but the baby died of an unrelated cause (cardiac problem).

The third case of death occurred in a 5-month-old baby with congenital heart disease receiving probiotics (*Bacillus clausii*) for watery diarrhoea; despite intensive antibiotics treatment, he finally succumbed to multidrug-resistant sepsis with multiorgan failure.

Three case reports [32,47,50], describing a total of four patients, suggested that a probiotic supplement taken by one hospital inpatient may spread to neighbouring patients, to whom it is not directly administered, leading to sepsis. In three cases, the etiological agent was confirmed by the identification, by molecular analysis, of the same probiotic strain compared with the probiotic administered to the neighbouring patient. In the other case, correlation was demonstrated by the isolation of the same microorganism in blood culture and CVC tip culture [50].

It has been suggested that contamination of vascular catheters may be responsible for such cases [49].

## 4. Discussion

This review summarised all the reported cases of paediatric patients who developed an invasive infection related to probiotics’ use. To our knowledge, this is the first review that focused specifically on the paediatric population. We were able to identify 49 documented paediatric cases of reported invasive infection caused by microorganisms used as probiotic supplementation.

Results of this review in terms of causative microorganisms (*Lactobacillus* spp. followed by *Saccharomyces* spp. and *Bifidobacterium* spp.) are slightly different from those reported in a previously published review in 2018 [25]. That study included 93 cases (both children and adults), identifying *Saccharomyces* spp. (50.5%) as the most frequent cause, while *Lactobacillus* spp. and *Bifidobacterium* spp. were present in 27.9% and 12.1% of cases, respectively. A higher frequency of fungemia in the adult population could explain this difference, since in that study, only 8 out of 34 patients with Saccharomyces-associated infection were children.

Previous reviews on the safety of probiotics have not found severe adverse events after the use of probiotics. In a review of 19 RCTs, including more than 2800 infants taking probiotics to prevent NEC, no cases of bacteraemia were reported, and the authors concluded that consumption of such products has a negligible risk to consumers [4]. Similar findings were reported by Borriello et al., who underlined the low risk in probiotics’ supplementation also in immunocompromised hosts [63]. In a retrospective study of two Italian neonatal units, no isolation of *Lactobacillus* species was reported in more than 5000 surveillance and clinical cultures [64]. The low incidence of severe adverse events could be explained by high standards of probiotics’ preparation, high standards of wards’ hygiene and care, and finally, by the fact that the great majority of children in these RCT’s did not have multiple comorbidities simultaneously.

We are aware that the incidence of probiotic-related infection cannot be compared between RCTs and case reports. Despite that, this review highlights how the risk of invasive infection during probiotic supplementation, although rare, should not be ignored, especially in patients with predisposing risk factors.

Findings on our review in terms of predisposing factors are largely in line with a previous systematic review [25]. Our review suggests that, in the paediatric population, prematurity is the major risk factor for developing a severe infection after probiotics’ use. The 2018 review [25] including adults showed that extreme ages are the most involved, with 35.5% of probiotics-associated infections occurring after 60 years and 26.7% in children younger than one year, of whom about 66% were premature. This could be explained by the systematic use of probiotics in premature children to prevent NEC (although trials on the NEC are mostly small, and some with high risk of bias, as stressed by the Cochrane review) and late-onset sepsis [4,5], but also by the susceptibility of their immature immune system to infections [65].

Similarly, intravenous catheters were identified as a frequent predisposing factor for developing invasive infections during probiotics’ use, also in the previous review [25]. In 23 cases (46.9%), patients had a pre-existing intestinal disease, such as short bowel syndrome, enteral/parental nutrition, intestinal inflammation, abdominal surgery, and diarrhoea, making intestinal comorbidity one of the most relevant predisposing factors for the development of probiotics-related infection. This could be explained by the extensive use of probiotics in patients with intestinal disorders, which may have an increased risk of probiotic translocation through the damaged intestinal mucosa. Several studies [27,49,66] suggest that a friable mucosa could potentially decrease adherence of the Lactobacillus and increase intestinal permeability, thereby potentiating migration of the organism across the intestinal mucosal barrier. Translocation may result in the transfer of bacteria to other organs, causing bacteraemia, septicaemia, and multiple organ failure [67].

Other high-risk groups included children with respiratory support (18.3%), congenital heart disease (12.2%), and genetic syndromes (8%), with less frequent underlying conditions being cystic fibrosis, malnutrition, burns, immunosuppressive treatment, chemotherapy for hematologic malignancies, and renal failure. Malignancy and immunosuppression related to HIV or immunosuppressive drugs were more common in the adult population [25]. More studies are needed to further elucidate the risk of invasive infections after probiotic use in these categories of patients.

Although most cases had favourable outcomes with appropriate antimicrobial therapy, children required hospitalisation and antimicrobial therapy. In addition, this review highlights three reported fatal cases, occurring in small children, of which two were premature babies taking probiotics according to a routine protocol for premature babies. In these cases, the cause of death was related to the underlying disease rather than to probiotic infection itself.

Limitations of this review include the number of databases searched. Future reviews may increase the data sources and complement the results of our reviews. Another limitation of our review is that we cannot exclude a publication bias, with other existing cases of probiotic-associated systemic infections not published in the scientific literature. Consequently, this review may underestimate the existing cases of probiotic-associated systemic infections in children. A further limit is that the review was not pre-registered on Prospero, but due to the nature of the study, it is unlikely that this may have resulted in a significant loss of data.

We cannot exclude that infections related to probiotics’ supplementation may have gone so far underestimated, due to other two reasons. First, probiotic products are regarded as dietary supplements in many countries, escaping the need to fall under regulatory frameworks for pharmaceutical products. There is usually no formal requirement to demonstrate safety and purity before marketing probiotics [68,69], and this can lead to significant inconsistencies between probiotic preparation stated in the product label and its actual content [70]. This may be even more relevant for products marketed outside the study setting, and may explain why in the study settings, where only highly standardised preparations are used, few adverse effects were reported. Second, probiotics are difficult to grow using standard culture media, so bacteraemia and fungemia from probiotic strains may be difficult to diagnose. This may have caused an underestimation of the actual incidence of severe infections due to probiotics. However, alerts of invasive infections related to probiotics are increasing, especially in high-risk groups such as preterm neonates and critically ill children [27,28,29,30,31,32,33,34,35,36,37,38,39,40,41,42,43,44,45,46,47,48,49,50,51,52,53,54,55,56,57,58,59,60,61,62]. In light of the accumulating evidence on probiotic-associated infections, we believe that any sepsis in patients undertaking probiotics, especially in patients with risk factors, should be evaluated for the specific probiotic strain in use.

Furthermore, some case reports [32,47,50] suggest that a probiotic supplement taken by one hospital inpatient may spread to neighbouring patients, through the contamination of vascular catheters [49]. In all these four reported cases, the strain-specific-isolated microorganism was the same one administered as a probiotic supplementation to the neighbouring inpatient. This suggests that poor infection control practices could have played a major role in causing probiotic infections in such cases. In this setting, the use of probiotics resulted as an additional risk factor for invasive infections in these patients. We suggest a careful handling of probiotic products in hospital settings to prevent possible shedding to other fragile patients.

Future trends of research could investigate the possible development of virulence in specific probiotic strains, using whole-genome sequencing, in order to improve the safety of these products.

## 5. Conclusions

While the effectiveness of probiotics is well-defined in few neonatal-specific conditions, their use is widespread in several other diseases, even in the absence of sound evidence of any clinically meaningful benefit. Findings of this review suggest that probiotics may be harmful in high-risk groups such as critically ill children with specific risk factors, such as prematurity, presence of intravenous catheters, pre-existing intestinal disease, presence of respiratory support, and congenital heart disease.

Given the large use of probiotics, especially in susceptible patients such as neonates, further studies aiming at evaluating the real incidence of probiotic-associated systemic infections in high-risk patients with predisposing factors are warranted.

## Figures and Tables

**Table 1 children-08-00924-t001:** Clinical and microbiologic features of 17 paediatric patients with *Lactobacillus* spp. (*L.* spp.) invasive infections after use of probiotics.

Ref.	Etiologic Agent *	Infection Type	Sex, Age	Underlying Condition(s)	Lenght of Probiotic Intake (Days)	Treatment (Days)	Outcome
[27]	*L. rhamnosus* GG	Sepsis	M, 4 months	Prematurity (36 weeks), short bowel syndrome, gastrostomy, cholestasis, chronic intestinal inflammation	23	cro+ amp (10)	Favourable
[27]	*L. rhamnosus* GG (ATCC53103)	Sepsis	M, 6 months	Prematurity (34 weeks), gastroschisis, short bowel syndrome, TPN, cholestasis, chronic intestinal inflammation	169	cro+ amp (10)	Favourable
[28]	*L. rhamnosus* GG (ATCC53103)	Sepsis	F, 6 years	Cerebral palsy, epilepsy, jejunostomy feeding, antibiotic-associated diarrhoea, CVC	44	NA (10)	Favourable
[28]	*L. rhamnosus* GG (ATCC53103)	Sepsis	M, 6 weeks	CHD, cardiac surgery, epilepsy, AKI, respiratory support, antibiotic-associated diarrhoea, CVC	20	pen G+ gen (24)	Favourable
[29]	*L. rhamnosus* GG (ATCC53103)	Sepsis	F, 3 months	Trisomy 18 and triple-X syndromes, CHD, respiratory support, CVC	88	cli (10)	Favourable
[29]	*L. rhamnosus* GG (ATCC53103)	Sepsis	M, 18 days	Prematurity (23 weeks), non-invasive respiratory support, CVC	16	gen (10)	Favourable
[30]	*L. rhamnosus* GG (ATCC53103)	Sepsis	M, 11 months	Prematurity (26 weeks), short bowel syndrome, cholestasis, cirrhosis, hypothyroidism, megaloblastic anaemia, CLD of infancy, CVC	35	amp+ gen (7)	Favourable
[31]	*L. rhamnosus* GG (ATCC53103)	Sepsis	M, 6 days	IUGR	4	tic+ca (14)	Favourable
[32]	*L. rhamnosus* GG (ATCC53103)	Sepsis	F, 18 days	Prematurity (25 weeks)	15	amp (17)	Favourable
[32]	*L. rhamnosus* GG (ATCC53103)	Sepsis	M, NA	Prematurity, CVC	neighbour	amp	Favourable
[32]	*L. rhamnosus* GG (ATCC53103)	Sepsis	M, NA	Prematurity, CVC	neighbour	amp	Favourable
[33]	*L. rhamnosus* GG (ATCC53103)	Bacteremia	M, 17 years	UC, concurrent enteric infection, immunosuppressive treatment, C. difficile colitis	5	tzp+ gen (5)	Favourable
[34]	*L. rhamnosus* GG (ATCC53103)	Sepsis	M, 2 months	Prematurity (25 weeks), spontaneous intestinal perforation, ileostomy	45	pen G (10)	Favourable
[35]	*L. rhamnosus* GG (ATCC53103)	Sepsis	F, 26 days	Prematurity (26 weeks), CVC	12	amp+ tzp (10)	Favourable
[36]	*L. rhamnosus* GG (ATCC53103)	Pneumonia	F, 11 months	Trisomy 21, esophageal surgery, gastrostomy, RSV infection	90	sam (10)	Favourable
[37]	*L. rhamnosus* GG (ATCC53103)	Sepsis	NA, 20 days	Prematurity (23 weeks)	19	NA	Favourable
[38]	*L. reuteri* (ATCC55730)	Sepsis	M, 2 days	Prematurity (27 weeks), respiratory support, UC	2	NA	death

NA = not applicable; TPN = Total Parental Nutrition; CLD = Chronic Lung Disease; CHD = Congenital Heart Disease; AKI = Acute Kidney Injury; UC = Ulcerative Colitis; IUGR = Intrauterine Growth Restriction; RSV= Respiratory Syncytial Virus; CVC: Central Venous Catheter; Cro = ceftriaxone; amp = ampicillin; cli = clindamycin; gen = gentamicin; pen = penicillin; tic = ticarcillin; ca = clavulanic acid; tzp = piperacillin/tazobactam; sam = ampicillin/sulbactam. * Strain was reported when available.

**Table 2 children-08-00924-t002:** Clinical and microbiologic features of 14 paediatric patients with *Saccharomyces* spp. (*S.* spp.) invasive infections after use of probiotics.

Ref.	Etiologic Agent	Infection Type	Sex, Age	Underlying Condition(s)	Lenght of Probiotic Intake (Days)	Treatment (Days)	Outcome
[39]	*S. boulardii*	Sepsis	M, 14 years	Burn, CVC	7	fc+ amB	favourable
[40]	*S. boulardii*	Fungemia	F, 1 year	Gastroenteritis, malnutrition, CVC	13	flz	favourable
[41]	*S. cerevisiae*	Sepsis	F, 17 days	Prematurity (26 weeks), CVC	1	NA	favourable
[42]	*S. boulardii*	Sepsis	M, NA	Prematurity (27 weeks), UVC, TPN	NA	mica	death
[42]	*S. boulardii*	Sepsis	M, NA	Prematurity (31 weeks), late-onset sepsis	NA	mica (14)	favourable
[43]	*S. cerevisiae*	Fungemia	NA, 8 months	AML, chemotherapy, neutropenia, CVC	1	L-amB (14)	favourable
[44]	*S. cerevisiae*	Sepsis	M, 3 weeks	Prematurity (30 weeks, IUGR)	4	NA (14)	favourable
[45]	*S. cerevisiae*	Sepsis	M, 3.5 months	Undiagnosed combined immunodeficiency	NA	amB (20)	favourable
[46]	*S. cerevisiae*	Sepsis	M, 8 years	Cerebral palsy, gastrostomy, aspiration pneumonia, CLD, CVC	NA	amB (14)	favourable
[47]	*S. cerevisiae*	Sepsis	F, 32 days	Esophageal atresia, tracheoesophageal fistula, CVC	neighbour	amB (21)	favourable
[48]	*S. cerevisiae*	Sepsis	M, 1 year	Trisomy 21, cardiac surgery, respiratory support, tracheostomy, dialysis catheter, malnutrition, CVC	4	amB (15)	favourable
[49]	*S. boulardii*	Sepsis	M, 30 months	Ileal atresia, small bowel resection, cystic fibrosis, malnutrition, TPN, CVC	300	amB (21)	favourable
[50]	*S. cerevisiae*	Fungemia	M, 3 months	CHD, TPN, CVC	10	L-amB	favourable
[50]	*S. cerevisiae*	Fungemia	F, 1 month	Intestinal atresia, small bowel resection, TPN, CVC	neighbour	NA	favourable

NA = not applicable; TPN = Total Parental Nutrition; CLD = Chronic Lung Disease; CHD = Congenital Heart Disease; AKI = Acute Kidney Injury; UC = Ulcerative Colitis; IUGR = Intrauterine Growth Restriction; UVC = Umbilical Venous Catheter; CVC: Central Venous Catheter; AML = Acute Myeloid Leukaemia; amB = amphotericin B; fc = flucytosin; flz = fluconazole; mica = micafungin.

**Table 3 children-08-00924-t003:** Clinical and microbiologic features of 15 paediatric patients with *Bifidobacterium* spp. (*B.* spp.) invasive infections after use of probiotics.

Ref.	Etiologic Agent *	Infection Type	Sex, Age	Underlying Condition(s)	Lenght of Probiotic Intake (Days)	Treatment (DAYs)	Outcome
[51]	*B. longum infantis* BIC 1206122787	Sepsis	F, 14 days	Prematurity (26 weeks), intussusception	9	caz+van (7), then ipm (7)	favourable
[51]	*B. longum infantis* BIC 140111125	Sepsis	F, 10 days	Prematurity (28 weeks), NEC	4	caz+amk+mtz	favourable
[52]	*B. longum*	Bacteriemia	F, 20 days	Prematurity (30 weeks), respiratory support, periumbilical infection (*Staphylococcus aureus)*	19	flx+gen (2)	favourable
[52]	*B. longum*	Bacteriemia	M, 20 days	Prematurity (28 weeks), respiratory support, BPD, UVC	14	amx+gen (2)	favourable
[52]	*B. longum*	Bacteriemia	F, 11 days	Prematurity (29 weeks), respiratory support, NEC	10	amc+gen	favourable
[53]	*B. longum infantis* ATCC15697	Sepsis	M, 2 weeks	Prematurity (23 weeks), spontaneous intestinal perforation	NA	ctx+gen	favourable
[53]	*B. longum infantis* ATCC15697	Sepsis	F, 5 weeks	Prematurity (24 weeks), NEC	NA	amp+gen+mtz	favourable
[53]	*B. longum infantis* ATCC15697	Bacteremia	M, 2 weeks	Prematurity (24 weeks)	NA	none	favourable
[54]	*B. longum infantis* ATCC15697	Sepsis	NA, 18 days	Prematurity (27 weeks)	9	ctx+van+mtz	favourable
[55]	*B. spp*	Bacteriemia	NA, 15 months	Heart disease (dilated cardiomyopathy, valvopathies, heart failure), ECMO, CVC	8	van+mem (7)	favourable
[56]	*B. breve* BBG-01	Bacteriemia	M, 8 days	Prematurity (36 weeks), cloacal exstrophy, omphalocele, imperforate anus, cystourethroplasty and colostomy, resection of the small intestine	8	cez+van	favourable
[57]	*B. breve*	Sepsis	M, 2 years	Acute lymphoblastic leukaemia, chemotherapy	NA	tpz+van+gen, then pen	favourable
[58]	*B. breve* BBG-01	Bacteriemia	F, 10 days	IUGR, abdominal surgery for omphalocele	NA	NA	favourable
[58]	*B. breve* BBG-01	Bacteriemia	M, 23 days	Preterm, Trisomy 21, Hirschsprung disease	NA	NA	favourable
[59]	*B. breve* BBG-01	Sepsis	M, 10 days	Prematurity (37 weeks), abdominal surgery for omphalocele, CVC	8	sam (2), amk (8), mem (10)	favourable

NA = not applicable, TPN = Total Parental Nutrition; BPD = Bronchopulmonary Dysplasia; CVC: Central Venous Catheter; UVC = Umbilical Venous Catheter; NEC = Necrotising Enterocolitis, IUGR = Intrauterine Growth Restriction; ECMO = Extracorporeal Membrane Oxygenation; caz = ceftazidime; van = vancomycin; ipm = imipenem; amk = amikacin; mtz = metronidazole; flx = flucloxacillin; amx = amoxicillin; amc = amoxicillin/clavulanate; ctx = cefotaxime; mem = meropenem; cez = cefazolina; lvx = levofloxacin; cst = colistin, sam = ampicillin/sulbactam. * Strain was reported when available.

**Table 4 children-08-00924-t004:** Clinical and microbiologic features of 3 paediatric patients with other probiotic invasive infections after use of probiotics.

Ref.	Etiologic Agent *	Infection Type	Sex, Age	Underlying Condition(s)	Lenght of Probiotic Intake (Days)	Treatment (DAYs)	Outcome
[60]	Bacillus clausii	Bacteremia	F, 17 months	No comorbidity, immunocompetent	4	amp, then lvx + gen, then van+ gen	favourable
[61]	Bacillus clausii	Sepsis	M, 5 months	Surgically corrected CHD	58	van (21) then mem+ cst	death
[62]	E. coli NISSLE 1917	Sepsis	NA 25 days	Prematurity (28 weeks)	10	mem+ van+ IVIg	favourable

NA = not applicable, CHD = Congenital Heart Disease; amp = ampicillin; gen = gentamicin; van = vancomycin; mem = meropenem; cez = cefazolina; lvx = levofloxacin; cst = colistin; IVIg = Intravenous immunoglobulin therapy. * Strain was reported when available.

## Data Availability

Data available in a publicly accessible repository.

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
