# Peer review of "Invasive Infections Associated with the Use of Probiotics in Children: A Systematic Review"

_children, 2021, doi:10.3390/children8100924_

Round 1
Reviewer 1 Report
This is an interesting topic and raising some important points around the lack of regulation surrounding the probiotic industry and the potential harm associated with this lack of regulation. It also raises the important point of ensuring clinical trials using probiotic supplements are reporting safety data as an outcome. For me, these are the 2 most valuable points gained from this data. The paper as it stands over-interprets the data and needs to be mindful of reporting the data as a case series only. There also needs to be more supporting evidence from the cases that it was the probiotic that caused the infections in these children and how each case reached these conclusions (method of testing etc). With the data as it is presented I am quite dubious about a number of the cases. In the end it is <50 cases in a 25-year period, which means that even if all this sepsis was caused by probiotics, probiotics are actually still remarkable safe considering their widespread use.
A few specific points:
INTRODUCTION:
Line 33-37: Streptococcus have also been isolated from the upper respiratory tract of healthy children. Not just the GIT.
Lines 37-42: Quite a biased interpretation of the literature. The evidence for systematic review for probiotics in the prevention of NEC is good, whilst there is little of no evidence for diarrhoea, allergic disease and infantile colic. I would omit phrases like “actual benefit measure by hard outcomes” and just report the facts and let the literature speak for itself.
Line 44-46 should be referenced. It is quite a broad sentence that isn’t then supported within the text of the paragraph
METHODS:
Reported as per PRISMA and the literature reviewed in duplicate. The protocol was not pre-registered on the PROSPERO database, however unlikely to have resulted in much bias in the outcomes for this review.
RESULTS:
- Were all the probiotics given orally? Enterally or by other means?
- How was the probiotic found to be the aetiological agent? Was it grown on blood culture? Strain-specific PCR? Co-incidence?
- I think the 4 cases that did not take probiotics supplement need to be examined in more detail. It is quite a leap to blame a probiotic administered to a neighbouring child. If the probiotic was a contaminant then the infection control practices surrounding the cases are likely to have allowed all sorts of pathogens and environmental contaminants to infect these children. Also, the investigations used to test for sepsis could have been equally contaminated and provided a false result.
- Sepsis is an umbrella term. What was the cause of sepsis in these children, was it bacteraemia?
- Why do you think CVC is a risk factor? Are treating nurses not practicing appropriate hygiene when accessing these CVCs and bacteria from the mouth or vomit are contaminating the lines? Again this is a red flag for poor infection control practices that would be much more concerning that probiotic use.
- I think I needs to be clear that of the 3 deaths, 2 were not caused by the probiotic.
TABLES:
The benefits from probiotics are strain specific, and I would anticipate their propensity to cause systemic infection would also be strain-specific. The strain, not just the species should be listed in the “etiologic agent”.
“Exposure time to probiotics” this is a confusing heading. I’m not sure what this means?
DISCUSSION:
Line 165: “Proven invasive infection caused by microorganisms used a probiotic supplement”. There is insufficient data within the manuscript to support this statement.
Line 174-175: I don’t think you can compare your specifically collected case series of adverse outcomes to reviews of RCTs as your aims are different to these studies, your data collection is different and hence your outcomes are different. It is important to include these studies in your discussion, but you cannot compare your results.
There is a good discussion of the limitations.
Line 232-236 is an important point both in the safety of probiotics, but also integrity and efficacy.
Line 273-239- also an important point!
244-246- also an excellent point.
Line 248- I think it may also reflect poor infection control practices within those institutions which is probably a very important cause of morbidity and mortality in immunocompromised infant.
As you have mentioned, the adverse outcomes appear to be more common outside of the well controlled clinical trails. Is it possible the some of these probiotics are becoming contaminated with virulent strains of well-known probiotics species, and whole genome sequencing of organisms isolated in “probiotic-sepsis” would be able provide information on how we can optimise the safety of these products?
Reviewer 2 Report
The article is in itself well written, but quite simplistic in its form and of questionable necessity. The case reports are simply listed, the statistics basic, and the tables need to be improved- there is no information regarding how the probiotic strain infection has been detected, how bacteremia and sepsis were distinguished, and missing data is variably marked (sometimes with N/A, sometimes with a -). There is no way of telling what the true incidence of the adverse effects registered in the case reports, and the review does not mention the potential detrimental effects of probiotic production not being monitored and regulated as closely as medication production. Moreover, the conclusion should be revised and more precisely define high-risk groups.
Round 2
Reviewer 1 Report
The authors have done an excellent job of addressing the suggestions.
Reviewer 2 Report
Dear authors, the changes and explanations which I noted were all considered and revised. After reviewing the new version of the manuscript, there are still a minor details which I would like to see changed, and that it the inclusion of the new name for Lactobacillus (Limosilactobacillus).